# Cadherin preserves cohesion across involuting tissues during *C. elegans* neurulation

**Kristopher M Barnes[1,2], Li Fan[1†], Mark W Moyle[3], Christopher A Brittin[1], Yichi Xu[1], Daniel A Colón-Ramos[3,4], Anthony Santella[1,5], Zhirong Bao[1]\***

[1]Developmental Biology Program, Memorial Sloan Kettering Cancer Center, New York, United States; [2]Graduate Program in Neuroscience, Weill Cornell Medicine, New York, United States; [3]Department of Neuroscience and Department of Cell Biology, Yale University School of Medicine, New Haven, United States; [4]Instituto de Neurobiología, Recinto de Ciencias Médicas, Universidad de Puerto Rico, San Juan, United States; [5]Molecular Cytology Core, Memorial Sloan Kettering Cancer Center, New York, United States

**\*For correspondence:**
baoz@mskcc.org

**Present address:** [†]Helen and Robert Appel Alzheimer's Disease Research Institute, Weill Cornell Medicine, New York, United States

**Competing interests:** The authors declare that no competing interests exist.

**Abstract** The internalization of the central nervous system, termed neurulation in vertebrates, is a critical step in embryogenesis. Open questions remain regarding how force propels coordinated tissue movement during the process, and little is known as to how internalization happens in invertebrates. We show that in *C. elegans* morphogenesis, apical constriction in the retracting pharynx drives involution of the adjacent neuroectoderm. HMR-1/cadherin mediates this process via inter-tissue attachment, as well as cohesion within the neuroectoderm. Our results demonstrate that HMR-1 is capable of mediating embryo-wide reorganization driven by a centrally located force generator, and indicate a non-canonical use of cadherin on the basal side of an epithelium that may apply to vertebrate neurulation. Additionally, we highlight shared morphology and gene expression in tissues driving involution, which suggests that neuroectoderm involution in *C. elegans* is potentially homologous with vertebrate neurulation and thus may help elucidate the evolutionary origin of the brain.

## Introduction

### Nervous system internalization

Neurulation is the process in chordates, which establishes the internalized position of the developing nervous system via involution of the neuroectoderm, with failure resulting in neural tube defects (NTDs) (*Gilbert and Michael J, 2019*). In tetrapods, neurulation involves an epithelial layer, which bends inwards until it is fully internalized and forms the neural tube (*Harrington et al., 2009*; *Nikolopoulou et al., 2017*), though in *Xenopus* this occurs simultaneously to lateral intercalation between the epithelial superficial cell layer and the deep cell layer (*Davidson and Keller, 1999*). In zebrafish, the deep cell layer is an epithelium-like layer that rolls up/involutes, while superficial cells migrate medially and intercalate into the deep cell layer to eventually form a monolayer (*Lowery and Sive, 2004*; *Hong and Brewster, 2006*; *Harrington et al., 2009*). More ancient clades including sharks and rays, hagfish, ascidians and amphioxus also show involution of a cohesive neural epithelium (*Lowery and Sive, 2004*), supporting a consistent role for epithelial bending/involution in neurulation.

Apical constriction produces force within the neural epithelium during neurulation, changing the cell shape to a wedge by shrinking the apical surface of the cell and causing tissue bending

(*Sawyer et al., 2010*). At the molecular level, force is generated via contraction of apical actomyosin, with cadherin localized to adherens junctions anchoring the contracting actomyosin and enabling the transmittance of force across the cell layer (*Ilina and Friedl, 2009*; *Martin and Goldstein, 2014*). In tetrapod neurulation, apical constriction is the strongest at the so-called medial and dorsolateral hingepoints of tissue bending (*Nikolopoulou et al., 2017*). In zebrafish, apical constriction has also recently been suggested to play a role in the initial involution despite the involuting layer not yet having acquired all the characteristics of a canonical epithelium (*Araya et al., 2019*). Subsequently, strong apical constriction occurs at the midline and lateral sides and forms the hingepoints (*Nyholm et al., 2009*). Perturbations which impair apical constriction, both genetic and by addition of cytochalasin D, result in cranial neural tube defects in mice (*Copp and Greene, 2010*; *Hildebrand and Soriano, 1999*; *Ybot-Gonzalez and Copp, 1999*; *Morriss-Kay and Tuckett, 1985*), *Xenopus* (*Haigo et al., 2003*; *Itoh et al., 2014*), and zebrafish (*Nyholm et al., 2009*; *Araya et al., 2019*). Additional cellular processes may contribute to neural tube formation, such as mesenchymal cell aggregation during secondary neurulation.

Coordination between adjacent tissues is an important aspect of neurulation and attachment between tissues may mediate the forces involved in morphogenesis (*Smith and Schoenwolf, 1997*). The attachment between the notochord and the neuroectoderm at the ventral midline is necessary to stabilize the involuting neuroectoderm (*Yang and Trasler, 1991*). Additionally, the connection between the epidermal ectoderm and the neuroectoderm may propel epidermal midline movement (*Smith and Schoenwolf, 1997*) but the nature of the connection varies significantly between species (*Harrington et al., 2009*). There remains much to be known as to how inter-tissue attachment is mediated during neurulation as well as its relative contribution to the process.

## *C. elegans* nervous system formation

The *C. elegans* nervous system has been mapped in its entirety (*Graham et al., 1986*), and the full and invariant cell lineage has been determined (*Sulston et al., 1983*), making it an ideal system for systems level study of embryonic nervous system morphogenesis. The central nervous system is composed of the nerve ring, the main neuropil composed of 181 axons, as well as the ventral nerve cord (VNC) (*Graham et al., 1986*). Most neurons in the embryo are born between 300 and 320 min post-fertilization (mpf) and proceed to internalize and move nearer to the midline of the embryo (*Harrell and Goldstein, 2011*), a process which has not been characterized in depth. Neurons subsequently begin projecting axons around early comma stage (~360 mpf) and proceed to form a visible ring within an hour (*Santella et al., 2015*; *Moyle et al., 2020*). During ventral cleft closure (*Chisholm and Hardin, 2005*), the initial movement of the VNC neurons to the midline and subsequent reorganization via PCP-mediated cell intercalation and convergent extension has been characterized (*George et al., 1998*; *Shah et al., 2017*). However, the mechanisms behind internalization of the neurons in the head, which will go on to form the nerve ring, have not been characterized.

These developmental events in the nervous system occur simultaneously with major morphogenetic events in non-neuronal tissues including the pharynx and hypodermis. The head nervous system develops alongside the pharynx (an organ unrelated to the vertebrate pharynx), and the nerve ring ultimately encircles it. Signaling from the pharynx during morphogenesis regulates the anterior-posterior placement of the nerve ring (*Kennerdell et al., 2009*). The bilayer pharyngeal primordium, located at the center of the nascent head through gastrulation (*Harrell and Goldstein, 2011*; *Pohl et al., 2012*), retracts into a bulb due to apical constriction during this time period (*Santella et al., 2010*; *Rasmussen et al., 2012*). This retraction is a major event in the course of head morphogenesis, but its link to the formation of the rest of the tissues in the head including that of the nervous system has not been studied.

The hypodermis forms on the dorsal exterior of the embryo during bean stage and subsequently extends to close and seal at the anterior and ventral midline. The ventral neuroblasts have been shown to be required for proper ventral closure (*George et al., 1998*), with hypodermis crawling over neuron substrates (*Wernike et al., 2016*). In head closure, it was recently demonstrated that anterior neuroblasts regulate the speed of hypodermal closure (*Grimbert et al., 2020*), and it is known that hypodermis closure requires the cadherin orthologue *hmr-1* (*Costa et al., 1998*).

In this paper, we connect the morphogenesis of the pharynx and hypodermis to the internalizing movement of the neurons. We show that this movement involves the involution of a cohesive neuroectoderm layer driven by attachment to the retracting pharynx in a pattern with striking similarity to

chordate neurulation, and we characterize the role of HMR-1 in establishing inter-tissue attachment and maintaining intra-tissue cohesion over the course of head formation.

## Results

### Nervous system involution is a coordinated process between the pharynx and the neuroectoderm

In order to characterize the process by which the nervous system internalizes in *C. elegans,* we examined head morphogenesis during the 60 min time window between the terminal division of the neurons and initial axon outgrowth. We first examined cell movements through WormGUIDES, a 4D atlas of *C. elegans* embryogenesis that tracks the position and lineage identity of every nucleus at every minute from the 4 cell stage to the one-and-half fold stage, about 2 hr after terminal division of neurons (*Santella et al., 2015*). The *C. elegans* neuroectoderm begins on the exterior of the embryo during early bean stage (~300 mpf). The pharyngeal cells have formed a two-sheet structure in the center of the head after invaginating during gastrulation. Head neurons envelop the pharynx at this stage (~300 mpf, *Figure 1a* first timepoint). One hour later, the pharynx has contracted into a bulb, and the neuroectoderm has moved to the anterior and ventral midline (*Figure 1a* second time-point, *Figure 1—Video 1*, *Figure 1—Video 2*), and effectively internalized.

To better examine cell shapes and potential cohesive relationships among cells during this process, we conducted 3D, time-lapse imaging with a pan membrane marker (*unc-33p::PH::GFP*) and a ubiquitous nuclear marker (histone::mCherry) to track the cell lineage (*Bao et al., 2006*; *Santella et al., 2014*; *Katzman et al., 2018*). In the transverse plane near the middle of the pharynx (*Figure 1b*), apical constriction of the two-sheet pharynx (*Figure 1b*, dashed circle), the inward/dorsal retraction of the pharynx, and the coordinated circumferential movement of neurons (*Figure 1b*, white dots and arrow) following the retracting pharynx are evident. Persistent cell contact is maintained between the neurons interfacing with the pharynx, as well as among the chain of involuting neurons. This coordinated movement occurs along the anterioposterior extent of the head. Visualized from the ventral side of the embryo (*Figure 1c*), movement trajectories of individual neurons (temporal max projections) show largely parallel tracts, which together with the persistent cell contacts observed above indicates tissue cohesion in the neuroectoderm and the pharynx during this movement (*Figure 1c*). Essentially all head neurons participate in this movement except for the amphid. During this process, the nascent hypodermis also moves anteriorly, with persistent cell contract between its leading edge and the trailing edge of the involuting neurons, before it eventually moves over the neurons to encase the head (*Figure 1—figure supplement 1*).

To explain the coordinated tissue movement and apparent tissue cohesion (*Figure 1d*), we propose a model in which retraction of the pharynx propels involution of the attached neuroectoderm, which is examined below.

### Involution of the head neurons requires *hmr-1/cadherin*

Because *hmr-1/cadherin* plays an important role in cell adhesion and is known to be widely expressed at this stage of embryogenesis (*Achilleos et al., 2010*), we asked if we can perturb neuroectoderm involution by inducing *hmr-1* lossof-function. To this end, we examined mutants homozygous -for the *zu248* loss-of-function allele given the previously demonstrated role of this allele in *C. elegans* head closure (*Costa et al., 1998*). We used a *cnd1p::PH::RFP* marker to label a set of bilateral ventral neurons (*Shah et al., 2017*) to measure their movements trajectories with live imaging. In the WT, these neurons migrate 25 microns over 60 min to meet at the midline (12/12). In *hmr-1* mutants, 75% of the embryos (9/12) fail to do so at the completion of pharynx retraction (*Figure 2a*), with an average of 28 microns separation between the bilateral neurons versus 0 microns for those in WT (*Figure 2b*). 25% (3/12) of *hmr-1* embryos successfully involute but arrest soon after as neurons appear to detach at the midline. We conclude that *hmr-1* is required for successful involution of the head neuroectoderm.

To rule out the possibility that the lack of neuroectoderm involution is due to a lack of pharynx retraction, where HMR-1 is known to localize to its apical side (*Sasidharan et al., 2018*), we measured the posterior movement of the anterior tip of the pharynx. Retraction is not significantly altered in any of the 12 *hmr-1* mutant embryos assessed (*Figure 2c*), with an average of 17 microns

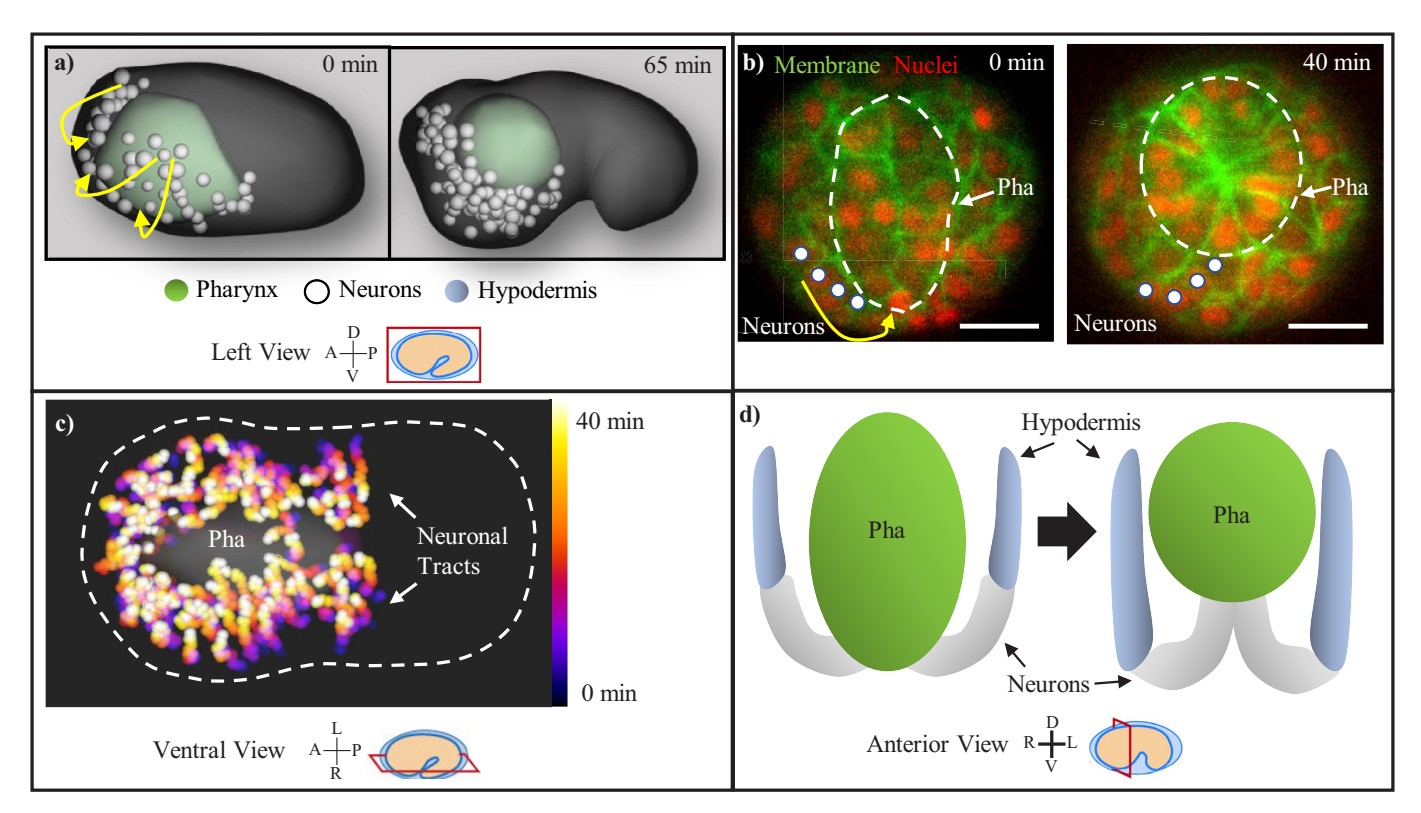

**Figure 1.** *C. elegans* nervous system centralization consists of the involution of the neuroectoderm with the retracting pharynx. (a) Head neurons (white) are observed enveloping the pharynx (green) as visualized in the WormGUIDES app. Yellow arrows indicate the circumferential path of the involuting neurons. Axis compass and view plane are displayed in this format throughout all figure panels. (b) Neuron chain (marked by white circles) during involution, shown in *unc33p::PH::GFP* expressing embryos (transverse plane at mid-pharynx, marked by white-dashed circle). Yellow arrow indicates path of involuting neurons. Red channel is cell nuclei. Scale bars are 10 µm. (c) Temporal max projection of involuting neurons on WormGUIDES, with chains progressing over 40 min from black/purple to yellow/white. White-dashed line is the embryo outline. (d) Model showing the pharynx (green), neuron chain (white), and hypodermis (blue) before and after pharynx retraction and involution.

The online version of this article includes the following video and figure supplement(s) for figure 1:

**Figure supplement 1.** Persistent contact between involuting neuroectoderm and hypoderm.

**Figure 1—video 1.** Video of involution in WormGUIDES embryo.

https://elifesciences.org/articles/58626#fig1video1

**Figure 1—video 2.** Anterior View, video of involution in WormGUIDES embryo.

https://elifesciences.org/articles/58626#fig1video2

**Figure 1—video 3.** Pan membrane (*Unc-33::PH::GFP*) imaging with histone:RFP marker, side view, shows leading edge of the hypodermis cohesively attached to involuting chain of neuroectoderm (marked by blue arrow).

https://elifesciences.org/articles/58626#fig1video3

involution in *hmr-1* mutants vs 18 microns for WT. This demonstrates that loss of neuroectoderm involution is independent of pharynx retraction in *hmr-1* mutant embryos and indicates additional function of *hmr-1* beyond apical constriction of the pharynx.

We then examined the effect of hmr-1 loss-of-function on the directed movement of individual neurons. To do so we measured the paths of a subset of neurons near the leading edge of the involution, namely SIBV, SIAD, AIY, and the mother of CEPV, on both the left and right side of the embryo based on live imaging and systematic cell lineage tracing (*Bao et al., 2006*; *Santella et al., 2014*; *Katzman et al., 2018*). These neurons at the leading edge of the involuting neuroectoderm display directed movement toward the midline, while in *hmr*-1 mutants they no longer move toward the midline and instead move slightly anterior (*Figure 2d*). Convergent movement of left-right homologs toward each other is significantly reduced in *hmr*-1 mutants in three out of four pairs

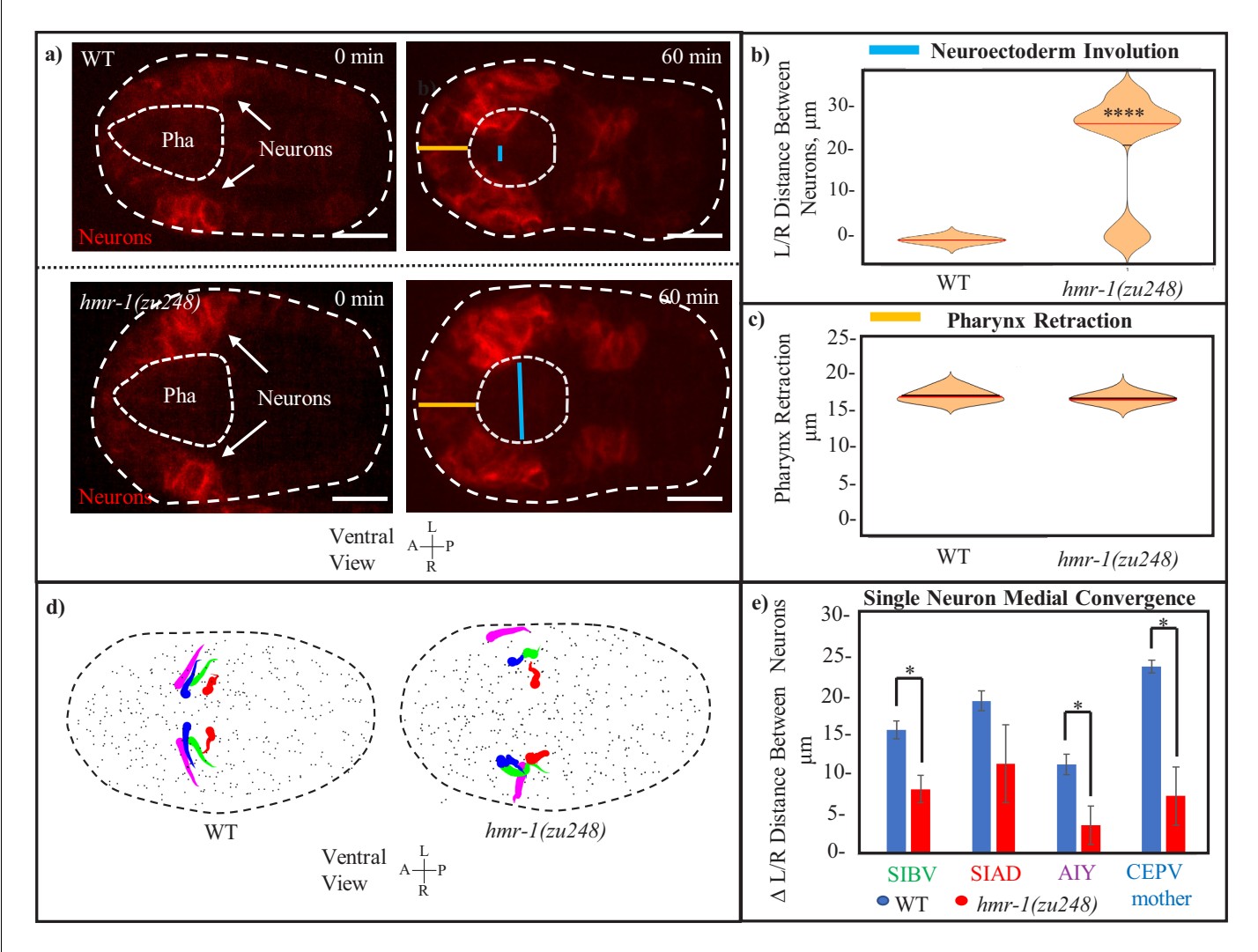

**Figure 2.** Involution of the *C. elegans* head neurons requires HMR-1. (a) Ventral neurons in *cnd1p::PH::RFP* expressing embryos move toward the midline during involution. Distance between left/right leading edges (blue line) is visibly increased in *hmr-1(zu248)* mutants. Outer dashed line is embryo outline, inner dashed line is pharynx. Distance of pharynx retraction (orange line) is conserved. Scale bars are 10 µm. (b) Involution of the neuroectoderm is measured as the distance between left and right side neurons in (n = 12) *hmr-1(zu248)* embryos versus in (n = 10) WT embryos (measurement of the blue line from panel a). Measurement is performed at the timepoint where the line is the shortest. Median is marked by red, significance was calculated with a one-tailed student's t-test. ****p < 0.001. (c) Pharynx retraction in WT vs *hmr-1(zu248)* embryos. Measurement is of length from anterior pole of embryo to anterior pole of pharynx (orange line in a). n = 6 embryos (WT) and n = 8 embryos (*hmr-1* mutant). Red line in plot marks median. (d) Motion paths of select neurons near the leading edge of involuting tissue in WT and *hmr-1(zu248)* embryos. Tadpole shape represents progression from early timepoints (tail) to late timepoints (head). Colors of neuron names in (d) are the same as the color of the equivalent motion paths (left and right). Dashed line is embryo outline, gray dots are other cells. (e) Quantification of movement trajectories from (c), measuring distance traveled by L/R partners toward each other. Significance across n = 3 embryos was calculated with a one-tailed student's t-test. *p < 0.05. The online version of this article includes the following source data for figure 2:

**Source data 1.** Measurements of neuroectoderm involution and pharynx retraction for *Figure 2b and c*, determined in Fiji from 10 *cnd1p::PH::RFP* embryos.

**Source data 2.** Data for *Figure 2e*.

measured, with between 40 and 70% less distance traveled in each (7.5–15 microns) (*Figure 2e*). These results reveal that the movement trajectories of neurons are shifted in *hmr-1* mutants.

## A localized HMR-1 patch at the pharynx/neuron interface

A key aspect of our model is the attachment of neuroectoderm to the retracting pharynx. Given the *hmr-1* phenotypes, we asked if HMR-1 is localized in such a way that it could mediate this attachment (*Figure 3a*). Indeed, a HMR-1::GFP reporter (*xnIs96*) which shows comparable localization patterns to the endogenous HMR-1 is strongly enriched at the interface between the basal pharyngeal surface and neuroectoderm during involution in a supracellular patch (*Figure 3b*, rectangle). This signal is distinct from the known localization at the apical side of the pharynx, which is dorsal to this patch (*Figure 3b*, arrowhead).

The existence of the HMR-1 patch at the neuron–pharynx interface is transient and coincides with the involution process (*Figure 3—Video 1*). Prior to pharynx retraction, HMR-1 expression across the embryo is low, prior to pharynx retraction, but localization to the interface and the formation of the supracellular patch are evident (*Figure 3b* first image). Notably, this patch forms before there is meaningful signal of HMR-1 localization at the apical side of the pharynx. HMR-1::GFP signal rises and stays high during pharynx retraction and involution (*Figure 3b* second and third images), and disappears afterwards (*Figure 3b* fourth image, with the remaining signal in the rectangle belonging to hypodermal cells). Quantification of HMR-1::GFP fluorescence intensity confirms this observation and further reveals the difference in temporal dynamics of HMR-1 localization between the supracellular patch at the neuron–pharynx interface and the apical side of the pharynx (*Figure 3c*), with the apical localization in the pharynx starting one step later and remaining high after involution completes.

Meanwhile, computational analysis of neighbor relationships between pharyngeal cells and the neurons adjacent to them further indicates where the attachment occurs at the physical interface. Specifically, we used Delaunay triangulation among nuclei to approximate neighbor relationship and detect cell pairs that persist as neighbors (see Methods, *Figure 3—figure supplement 1a*). Using data from 3 WT embryos where the entire cell lineage was traced up to the one-and-half fold stage (*Santella et al., 2015*), we identified neurons that maintain their original pharyngeal neighbors during >75% of timepoints during involution (*Figure 3—figure supplement 1b*, *Table 1*). This analysis identified two groups of neurons, one on the anterior and one on the ventral side of the pharynx (*Figure 3d*). The ventral group (*Figure 3d*, yellow circle), which includes SMDD, RIS and RMEV, spatially coincides with the supracellular HMR-1 patch, supporting the concept that HMR-1 promotes cohesion across the tissue interface between the neurons and the basal pharyngeal surface. We were not able to examine HMR-1 localization at the anterior of the pharynx due to technical difficulties in orienting the tilting pharyngeal surface to the imaging axis.

We then examined if *hmr-1* is required for the correlated movement between the interface neurons and the adjacent pharyngeal cells. We measured this using the displacement of cells in WT and *hmr-1* mutant embryos (*Figure 3e*). Four of the interface cells, namely SMDDL, SMDDR, RIS, and RMEV, show coordinated motion with their pharyngeal neighbors in WT embryos. In *hmr-1* mutants, their movement trajectories diverge from their initial pharyngeal neighbors. These neurons move significantly less in *hmr-1* mutants compared to the WT, with an average of 38% (15 microns) less displacement measured across three embryos, while pharynx displacement is not significantly changed (*Figure 3f*). Based on the localization, phenotypes, and computational analysis, we conclude that HMR-1 mediates the attachment between the interface neurons and the pharynx, though further tissue-specific mutant studies would be required to definitively connect this phenotype to loss of the HMR-1 patch at the interface.

## Cohesion within the neuroectoderm requires HMR-1

Furthermore, we examined whether HMR-1 is also required for cohesion within the neuroectoderm by examining its localization between neurons, and whether its loss would result in sliding between neighboring neurons (*Figure 4a*). HMR-1 localizes between involuting neurons with a stronger signal than that in non-involuting neurons in the amphid (*Figure 4b,c*, *Figure 4—figure supplement 1*). To assay potential sliding, we examined the correlation of movement trajectory among neurons. Compared to the WT, neurons in *hmr-1* mutants show reduced, less directional movement toward the midline (*Figure 4d*). We further quantified the correlation between individual movement trajectories within a select group of neighboring neurons (SIBV, SIAD, AIY, and the mother of CEPV on the left and right side) (*Figure 4e*). In WT embryos, their respective movement is positively correlated on

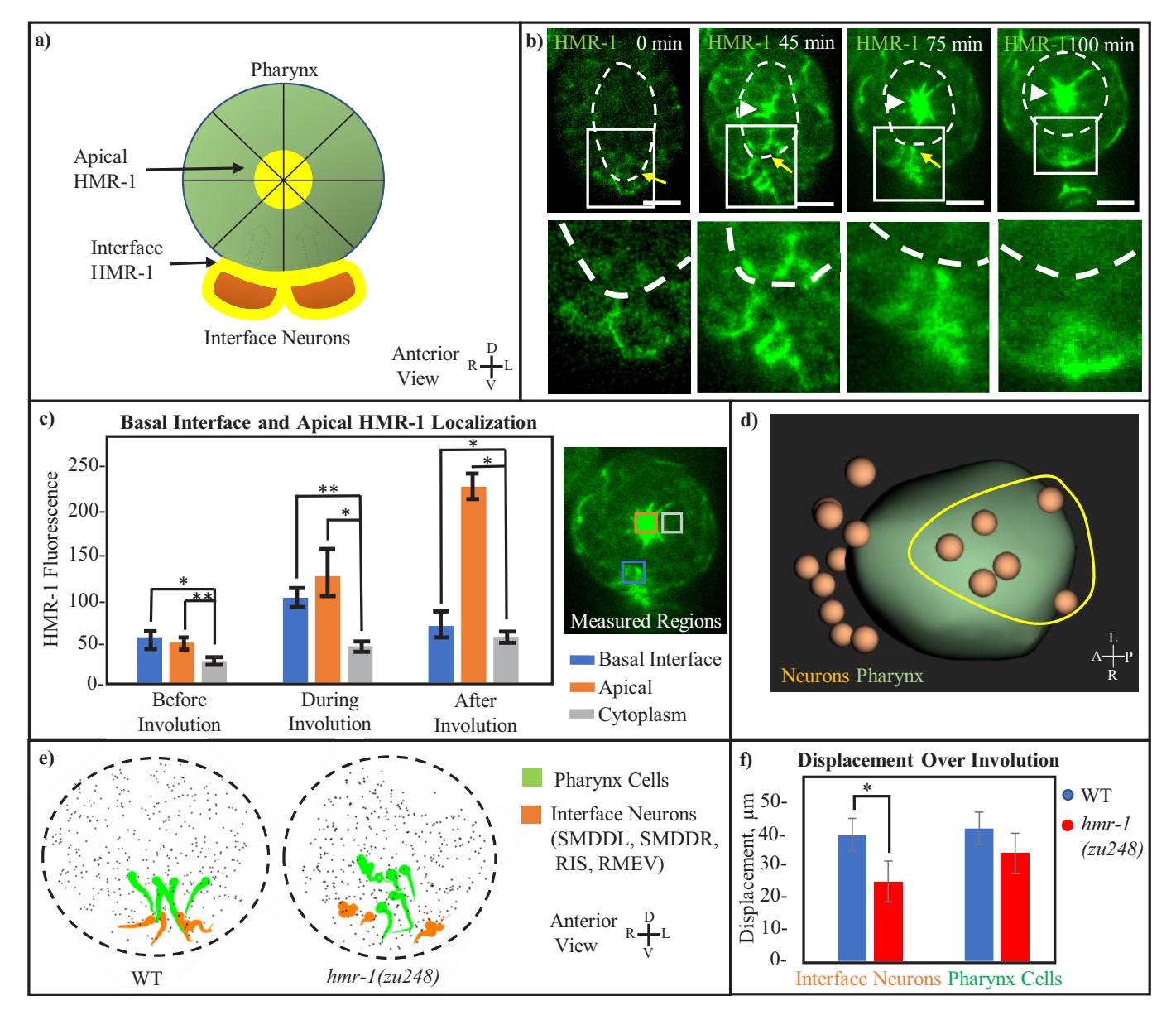

**Figure 3.** Cohesion at the inter-tissue interface is maintained by a local HMR-1 patch. (a) Model of the adhesive HMR-1 interface (yellow) at the connection between ventral interface neurons (orange) and the retracting pharynx (green) mediating inter-tissue cohesion during involution. Apical HMR-1 is labeled, and lines through pharynx indicate apical constriction. (b) Anterior view HMR-1::GFP timeseries showing the basolateral (interface) and apical HMR-1 patches, starting before pharynx retraction when the interface patch is first forming, and ending after pharynx retraction when the interface patch disappears and only the apical localization remains. White arrowheads indicate apical HMR-1, yellow arrow indicates interface HMR-1. Cutouts show enlarged view of the interface at each timepoint. White-dashed line indicates pharynx. Scale bars are 10 µm. (c) Local HMR-1::GFP fluorescence intensity minus background, assessed across n = 5 embryos at three timepoints before, during, and after retraction. Measurements were taken in an anterior view, with measured regions highlighted in corresponding colors. Error bars represent standard deviation at given timepoint. A 2-tailed t-test with equal variance was performed, * < 0.05, **p < 0.01. (d) Interface neurons, defined as those with a persistent connection to a pharynx cell from the beginning of involution for at least 75% of timepoints, form a patch near the ventral midline (full data in *Figure 3—figure supplement 1* and *Table 1*). Yellow circle indicated rough shape of the group of interface neurons on the ventral side of the pharynx. (e) Anterior view shows motion paths of four interface neurons (orange) as well as their pharyngeal neighbors (green) in WT and *hmr-1(zu248)*. Tadpoles are as in (2e). Black dashed circle is embryo outline, gray dots are other cells. (f) Graph showing the total displacement over the timecourse of involution (n = 3 embryos). Neurons: SMDDL, SMDDR, Pharynx Cells: Selected in each embryo according to proximity to SMDDL/R. *p < 0.05.

The online version of this article includes the following video, source data, and figure supplement(s) for figure 3:

**Source data 1.** Data for *Figure 3c*.

*Figure 3 continued on next page*

*Figure 3 continued*

**Source data 2.** Data from *Figure 3f*.

**Figure supplement 1.** Computational assessment of persistent cohesion from relative nuclear location.

**Figure 3—video 1.** HMR-1:GFP localization at the ventral pharynx–neuron interface in addition to the apical lumen of the pharynx.

https://elifesciences.org/articles/58626#fig3video1

**Table 1.** Persistence of pharynx contact across neurons and selected leader cells.

| Cell | Timepoints contacting initial neighbors | % Time | Cell | Timepoints contacting initial neighbors | % Time | Cell | Timepoints contacting initial neighbors | % Time |
|---|---|---|---|---|---|---|---|---|
| 'AIAL' | 0 | 0.0 | 'IL2DR' | 27 | 41.5 | 'RMGR' | 0 | 0.0 |
| 'AIAR' | 0 | 0.0 | 'IL2L' | 0 | 0.0 | 'SAADL' | 0 | 0.0 |
| 'AIML' | 0 | 0.0 | 'IL2R' | 0 | 0.0 | 'SAADR' | 7 | 10.8 |
| 'AIMR' | 0 | 0.0 | 'IL2VL' | 0 | 0.0 | 'SAAVL' | 43 | 66.2 |
| 'AINL' | 35 | 53.8 | 'IL2VR' | 0 | 0.0 | 'SAAVR' | 18 | 27.7 |
| 'AINR' | 56 | 86.2 | 'OLLL' | 0 | 0.0 | 'SABD' | 0 | 0.0 |
| 'AIYL' | 39 | 60.0 | 'OLLR' | 0 | 0.0 | 'SABVL' | 0 | 0.0 |
| 'AIYR' | 40 | 61.5 | 'OLQDL' | 63 | 96.9 | 'SABVR' | 0 | 0.0 |
| 'ALA' | 0 | 0.0 | 'OLQDR' | 65 | 100.0 | 'SIADL' | 7 | 10.8 |
| 'AVAL' | 0 | 0.0 | 'OLQVL' | 0 | 0.0 | 'SIADR' | 23 | 35.4 |
| 'AVAR' | 0 | 0.0 | 'OLQVR' | 0 | 0.0 | 'SIAVL' | 51 | 78.5 |
| 'AVDL' | 65 | 100.0 | 'RIAL' | 0 | 0.0 | 'SIAVR' | 18 | 27.7 |
| 'AVDR' | 34 | 52.3 | 'RIAR' | 32 | 49.2 | 'SIBVL' | 0 | 0.0 |
| 'AVEL' | 0 | 0.0 | 'RID' | 16 | 24.6 | 'SIBVR' | 2 | 3.1 |
| 'AVER' | 0 | 0.0 | 'RIFL' | 0 | 0.0 | 'SMBDL' | 11 | 16.9 |
| 'AVG' | 51 | 78.5 | 'RIFR' | 0 | 0.0 | 'SMBDR' | 17 | 26.2 |
| 'AVHL' | 65 | 100.0 | 'RIGL' | 0 | 0.0 | 'SMBVL' | 0 | 0.0 |
| 'AVHR' | 54 | 83.1 | 'RIGR' | 0 | 0.0 | 'SMBVR' | 0 | 0.0 |
| 'AVJL' | 0 | 0.0 | 'RIH' | 27 | 41.5 | 'SMDDL' | 65 | 100.0 |
| 'AVJR' | 0 | 0.0 | 'RIPL' | 54 | 83.1 | 'SMDDR' | 53 | 81.5 |
| 'AVKL' | 37 | 56.9 | 'RIPR' | 0 | 0.0 | 'SMDVL' | 52 | 80.0 |
| 'AVKR' | 0 | 0.0 | 'RIR' | 41 | 63.1 | 'SMDVR' | 10 | 15.4 |
| 'AVL' | 33 | 50.8 | 'RIS' | 59 | 90.8 | 'URAVL' | 0 | 0.0 |
| 'BAGL' | 32 | 49.2 | 'RIVL' | 65 | 100.0 | 'URAVR' | 0 | 0.0 |
| 'BAGR' | 42 | 64.6 | 'RIVR' | 0 | 0.0 | 'URADL' | 0 | 0.0 |
| 'CEPDL' | 26 | 40.0 | 'RMDDL' | 22 | 33.8 | 'URADR' | 0 | 0.0 |
| 'CEPDR' | 31 | 47.7 | 'RMDDR' | 16 | 24.6 | 'URBL' | 0 | 0.0 |
| 'CEPVL' | 0 | 0.0 | 'RMDL' | 0 | 0.0 | 'URBR' | 0 | 0.0 |
| 'CEPVR' | 0 | 0.0 | 'RMDR' | 0 | 0.0 | 'URXL' | 26 | 40.0 |
| 'IL1DL' | 19 | 29.2 | 'RMDVL' | 0 | 0.0 | 'URXR' | 31 | 47.7 |
| 'IL1DR' | 19 | 29.2 | 'RMDVR' | 18 | 27.7 | 'URYDL' | 13 | 20.0 |
| 'IL1L' | 0 | 0.0 | 'RMED' | 16 | 24.6 | 'URYDR' | 44 | 67.7 |
| 'IL1R' | 42 | 64.6 | 'RMEL' | 41 | 63.1 | 'URYVL' | 0 | 0.0 |
| 'IL1VL' | 0 | 0.0 | 'RMER' | 54 | 83.1 | 'URYVR' | 0 | 0.0 |
| 'IL1VR' | 0 | 0.0 | 'RMEV' | 65 | 100.0 | | | |
| 'IL2DL' | 36 | 55.4 | 'RMGL' | 0 | 0.0 | | | |

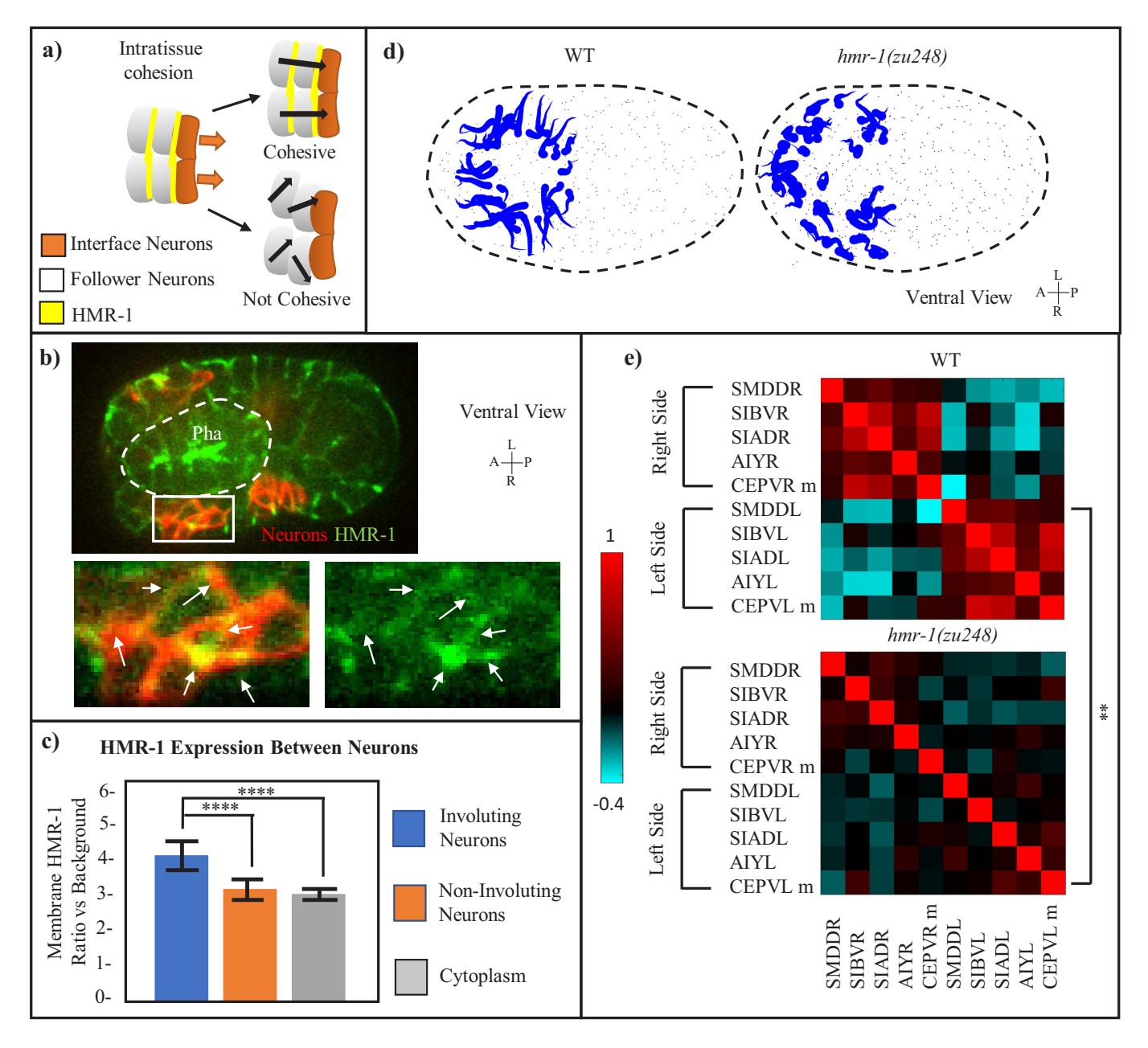

**Figure 4.** Intratissue Cohesion in the Neuroectoderm also Requires *hmr-1*. (**a**) Model of expected neuronal behavior with and without intratissue cohesion. Yellow lines indicate adhesive interfaces between cells. Orange arrows mark the movement of interface neurons with the pharynx, black arrows mark the expected path of neurons. (**b**) HMR-1::GFP localization at the connection between involuting neurons. The patch of neurons is marked by *cnd1p*::PH::RFP expression and enlarged in cutout. White-dashed line represents pharynx. White arrows in cutout indicate HMR-1::GFP signal at cell boundaries. Scale bars = 10 μm. (**c**) Quantification of HMR-1::GFP fluorescence at involuting neuronal boundaries, assessed at *cnd1p::PH::RFP*+ membranes (n = 6 embryos, five membranes per embryo). Amphid neurons, which are not involved in involution, are used as a control. Error bars represent standard deviation. Statistical comparison was done with a 2-tailed t-test for equal variance, ****p < 0.0001. (**d**) Motion paths of neurons in WT and *hmr-1(zu248)* embryos, tadpoles are as in (2e). Black dashed circle is embryo outline, gray dots are other cells. (**e**) Movement path correlation of select involuting neurons in a cluster on the ventral side, between WT (n = 3) and *hmr-1(zu248)* (n = 3) embryos. Bar indicates correlation value for each color (1 = moving in same direction, −1 = moving in opposite directions). Left side vs right side neurons are labeled. P-value of. 01 calculated with paired t-test applied globally between neurons. M is mother. **p < 0.01.

The online version of this article includes the following source data and figure supplement(s) for figure 4:

**Source data 1.** Data for *Figure 4c*.
**Source data 2.** Data for *Figure 4e*.
*Figure 4 continued on next page*

*Figure 4 continued*

**Figure supplement 1.** Amphid Neurons Fail to Move toward the Ventral Midline.

the ipsilateral side, and anticorrelated with the contralateral side. In contrast, *hmr-1* mutants show a general lack of correlation regardless of sides. These data were normalized across three WT and three mutant embryos. Together, the localization and phenotype assays support the hypothesis that HMR-1 also mediates intratissue cohesion within the neuroectoderm.

### Similarity between *C. elegans* head involution and chordate neurulation

We found considerable similarity between the coordinated tissue movement in *C. elegans* head involution versus chordate neurulation (*Figure 5a*). Both share the topology defined by medial involution of a cohesive neuroectoderm layer with an underlying mesodermal organ at the ventral midline and flanked by the hypodermal/epidermal ectoderm laterally. In vertebrates, the floor plate generates force through apical constriction at the medial hingepoint to propel involution and the neuroectoderm layer is attached to the notochord to stabilize internalization (*Yang and Trasler, 1991*), while in *C. elegans* the pharynx accomplishes both these roles. This in turn helps propel the midline closure of the hypodermis/epidermis (*Smith and Schoenwolf, 1997*; *Grimbert et al., 2020*). One significant difference is that involution and hypodermis/epidermis closure happen on the dorsal side in chordates but the ventral side in *C. elegans*. However, it is established that a flip of the D-V axis occurred in deuterostome evolution (*Arendt and Nübler-Jung, 1994*).

A more striking similarity is the underlying developmental program of the floor plate and notochord in vertebrate compared to the pharynx in *C. elegans*. The floor plate does not give rise to neurons, and lineage wise is in-part derived from the midline precursor cells (MPCs) in the organizer/node region along with the notochord. (*Teillet et al., 1998*; *Le Douarin and Halpern, 2000*; *Peyrot et al., 2011*). That is, as the pharynx in *C. elegans*, the MPC derived floor plate, and the notochord are extrinsic to the neuroectoderm. Furthermore, as the pharynx in *C. elegans*, the cells giving rise to the floor plate and notochord express *pha-4/foxa2* (*Horner et al., 1998*; *Teillet et al., 1998*; *Jeong and Epstein, 2003*). In chick, additional *pha-4/foxa2+* cells contribute to the anterior floor plate (*Patten et al., 2003*). We discuss the potential evolutionary meaning of this similarity below.

## Discussion

### Role of cadherins in *C. elegans* nervous system involution

Our work shows that HMR-1/cadherin plays a critical role in coordinating the multi-tissue process to internalize the nervous system, both in terms of inter-tissue attachment and cohesion within the neural tissue. Similarly, loss of cadherins causes defects in vertebrate neurulation in zebrafish (*Lele et al., 2002*; *Hong and Brewster, 2006*; *Araya et al., 2019*), *Xenopus* (*Nandadasa et al., 2009*) and rat (*Chen and Hales, 1995*). Given the similarity of the processes in *C. elegans* and vertebrates that we have identified and the relative simplicity, *C. elegans* provides a useful model to further elucidate the biomechanics of tissue movement and remodeling in neurulation and the role of cadherins.

Cadherins are well known for their established roles at the apical membrane during apical constriction, and for their presence on the lateral membrane to maintain cohesion within an epithelium. However, our work shows HMR-1 mediates inter-tissue adhesion on the basal side of the pharynx. If HMR-1 is in fact on the pharynx basal membrane it would raise an interesting question as to how a cell coordinates two distinct locations. On the other hand, if HMR-1 is only used in the interface neurons, it raises the question of what functions on the pharynx side to bind to HMR-1. It is worth noting that in vertebrates, functional attachment between the floor plate and the notochord (both of which are epithelial) occurs between their basal sides (*Smith and Schoenwolf, 1997*). Upon closer examination of published data, cadherin is transiently localized to the basal side of the floor plate/neuroectoderm and notochord (*Dady et al., 2012*; *Figure 4*). Tissue-specific labeling and perturbation of HMR-1 will start to address these questions in *C. elegans* and should motivate similar efforts in vertebrates.

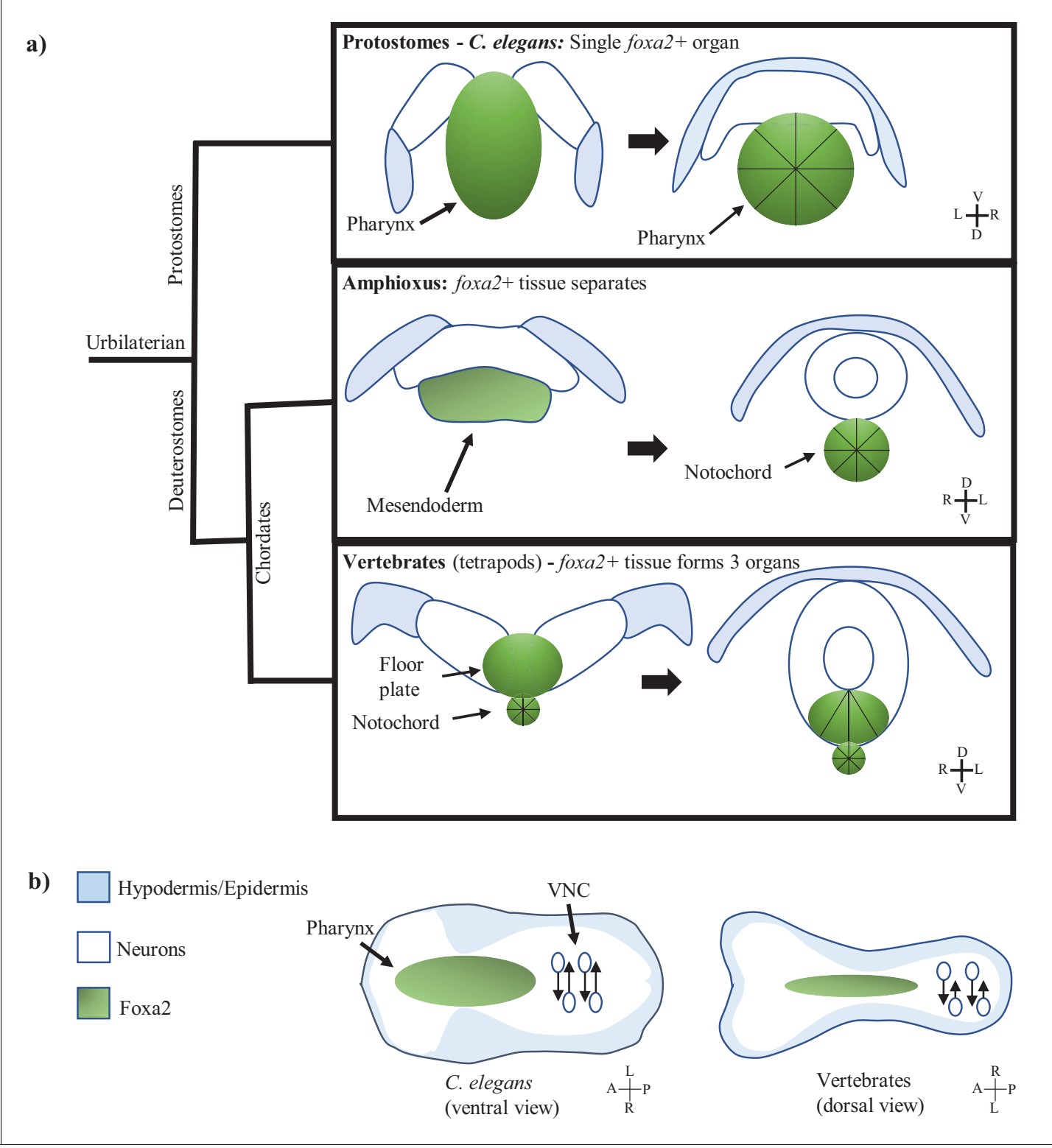

**Figure 5.** The multi-tissue process of *C. elegans* involution may be homologous to chordate neurulation. (**a**) Model showing *foxa2* specified force generator during nervous system involution across three clades. Left images are before involution, right images are after. Black lines indicate apically constricting cells. Protostomes (*C. elegans*) have involution driven by a single *foxa2*+ tissue (the pharynx). Amphioxus also has a single *foxa2*+ layer from which the future notochord forms. Vertebrates have separation of the *foxa2*+ tissue into the floor plate and notochord before involution. (**b**) Model of morphology during nervous system involution in *C. elegans* as well as chordates (an approximation based off of amniote neurulation), labeling

*Figure 5 continued on next page*

*Figure 5 continued*

involution in the anterior of the embryo (pharynx in green, neuroectoderm as white background) and PCP/CE-driven nerve cord formation in the posterior of the embryo (neurons as white circles). Hypodermis is in blue. Body axis is flipped between *C. elegans* and vertebrates (in (b) as well).

## Potential evolutionary homology and implications

While neurons originated in radially symmetric animal phyla such as Ctenophora and Cnidaria, they began to assemble into complex centralized systems such as nerve cords and brains only in Bilateria. The traditional view holds that central nervous systems evolved multiple times, but this idea has been challenged recently. Multiple studies have suggested that the brain only evolved once in the ancestral bilaterian (the 'urbilaterian theory' of brain evolution). These arguments are based on similarities in anatomy and gene expression between clades (*Hirth et al., 2003*; *Magie et al., 2005*; *Denes et al., 2007*; *Watanabe et al., 2009*; *Bailly et al., 2013*; *Holland et al., 2013*; *Arendt et al., 2016*).

The term neurulation has been used loosely in *C. elegans* to refer to various aspects of neural morphogenesis such as the closure of the ventral neuroblasts (*Wadsworth et al., 1996*) or the gastrulation of the ventral ectoderm (*Harrell and Goldstein, 2011*). However, according to the model that the brain evolved once in the ancestral bilaterian, the striking similarity between nervous system internalization in *C. elegans* and vertebrates could imply that the process in *C. elegans* is homologous to bonafide neurulation (and can be referred to as such). In return, it would further strengthen the current arguments for the urbilaterian brain theory (which are largely based on the molecular mechanism of lineage differentiation and neural fate specification) by providing shared functional cell biological aspects in morphogenesis, as well as shared essential gene expression in tissues involved in coordinated morphogenesis beyond the nervous system itself. Since these aspects have been well characterized in chordates, which are deuterostomes, the strongest evidence would come from a protostome, which *C. elegans* represents.

Extensive investigations are needed in both deuterostomes and protostomes to fully examine if a homologous process of neurulation exists in the latter. On the side of deuterostomes, one needs to better define the cell biological mechanisms of neurulation in more primitive chordates such as amphioxus (*Figure 5a*) and hemichordates to define the force generators and their lineage origin. Furthermore, as mentioned above, the relevant structures in vertebrates, namely the floor plate and the notochord, are derived from the *pha-4/foxa2+* precursor cells in the organizer region, along with another structure called the dorsal endoderm (*Teillet et al., 1998*). One possibility is that the cell types in the organizer region are homologous to or evolved from the force generators in protostomes and became more elaborated in structure to give rise to the floor plate and notochord. Consistent with this notion, the mesendodermal tissue that gives rise to the notochord in amphioxus is *pha-4/foxa2+* and appear to undergo apical constriction (*Albuixech-Crespo et al., 2017*; *Holland et al., 1996*). It has been proposed that the notochord evolved from the mesendoderm in hemichordates (*Annona et al., 2015*). The above reasoning would suggest that we may be able to trace the evolutionary origins of neurulation even further by examining the force generators in protostomes.

On the side of protostomes, our work makes *C. elegans* the first system where the morphogenetic mechanism is defined for the internalization of the nervous system. Additional species need to be examined to show whether the general scheme is shared among protostomes. Preliminary studies in *D. melanogaster*, where the neuroectoderm moves to the midline after an involuting mesoderm layer (*Mizutani and Bier, 2008*), make this species a potentially productive choice.

Lastly, the morphogenesis of the VNC of *C. elegans,* as characterized in our previous study, also draws significant parallels to chordate neurulation (*Figure 5b*). Both processes are based on PCP-mediated mediolateral cell intercalation and convergent extension to elongate the nervous system along the A-P axis (*Elul et al., 1997*; *Williams et al., 2014*; *Shah et al., 2017*). This additional morphogenetic mechanism shared between *C. elegans* and chordates makes it more difficult to argue for convergent evolution. Overall, our findings provide a strong argument from the perspective of shared developmental morphology and a conserved force generator to favor the hypothesis that the brain only evolved once.

# Materials and methods

## Key resources table

| Reagent type (species) or resource | Designation | Source or reference | Identifiers | Additional information |
|---|---|---|---|---|
| Genetic reagent (*E. coli*) | OP50 | Caenorhabditis Genetics Center | OP50 | RRID:WB-STRAIN:WB Strain 00041971 |
| Genetic reagent (*C. elegans*) | xnIs96 ([hmr-1p::hmr-1::GFP::unc-54 3'UTR + unc-119(+)]) | Caenorhabditis Genetics Center | FT250 | RRID:WB-STRAIN:WB Strain 00007535 |
| Genetic reagent (*C. elegans*) | zyIs36 [cnd1-p::PH::RFP]X | Dr. Antonio Colavita | OU412 | |
| Genetic reagent (*C. elegans*) | olaex2540 [unc-33p:PH:GFP] | Dr. Daniel Colon-Ramos | DCR4318 | |
| Genetic reagent (*C. elegans*) | ujIs113[pie-1::mCherry::H2B + unc-119(+); Pnhr-2::mCherry::histone + unc-119(+)] II | Caenorhabditis Genetics Center | JIM113 | RRID:WB-STRAIN:WB Strain 00022462 |
| Genetic reagent (*C. elegans*) | hmr-1(zu248) | Caenorhabditis Genetics Center | JJ1142 | RRID:WB-STRAIN:WB Strain 00022484 |
| Genetic reagent (*C. elegans*) | zuEx2 (W02B9(cosmid) + rol-6(su1006)) | Caenorhabditis Genetics Center | JJ1142 | RRID:WB-STRAIN:WB Strain 00022484 |
| Software, algorithm | Fiji | https://fiji.sc/ | | |
| Software, algorithm | MATLAB | https://www.mathworks.com | | |
| Software, algorithm | Metamorph | https://www.moleculardevices.com | | |
| Software, algorithm | Starry Nite | https://wormguides.org/starry-nite/ | | |
| Software, algorithm | AceTree | https://github.com/zhirongbaolab/AceTree | | |
| Software, algorithm | WormGUIDES atlas | https://wormguides.org/wormguides-atlas/ | | |

## *C. elegans* strains and genetics

*C. elegans* strains were grown on NGM plates seeded with OP50 bacteria as previously detailed (*Brenner, 1974*). N2 Bristol was used as the WT strain. All worms were grown at room temperature. The following strains were used throughout the study: BV292(zyIs36[cnd-1p::PH::RFP] IV), BV308 (zyIs36 [cnd1-p::PH::RFP]X, unc-119(ed3) III; xnIs96 [hmr-1p::hmr-1::GFP::unc-54 3'UTR + unc-119 (+)]), BV727 (ujIs113, hmr-1(zu248) I; zuEx2), DCR4318 (olaex2540 [Punc-33_PHD_GFP_unc54, Punc-122_RFP]; ujIs113), BV745(zyIs36[cnd-1p::PH::RFP] IV, hmr-1(zu248) I; zuEx2).

## Embryonic imaging

Preparation of embryos for live imaging was done as previously described (*Bao and Murray, 2011*). Gravid adult worms were picked to ~30 µl M9 buffer (3 g KH$_2$PO$_4$, 6 g Na$_2$HPO$_4$, 5 g NaCl, 1 ml 1 M MgSO$_4$, per liter H$_2$O), transferred to a second drop to dilute extra OP50 bacteria, and cut open to release embryos. Depending on the goal of the imaging experiment, one of three next steps would be taken. When embryos were to have their lineage analyzed using AceTree software, 4–10 embryos at 2–4 cell stage were transferred to a small drop (~1.5 µl) of M9 media mixed with 20 µm polystyrene beads on a 24 × 50 mm coverslip in and sealed with vaseline under an 18 × 18 mm smaller coverslip. When embryos were intended only for visual phenotyping, the same protocol was used, but up to 40 embryos were imaged at once. Lastly, when embryos were to be viewed through an anterior-posterior orientation, embryos were added to a larger (~4 µl) drop of M9 without beads, a thin layer of vaseline was deposited above and below the drop, and the 18 × 18 mm coverslip was added on top and further sealed with an extra layer of vaseline to allow uncompressed imaging.

Images were acquired on either a spinning-disk confocal microscope comprising a Zeiss Axio Observer Z1 frame with an Olympus UPLSAPO 60XS objective, a Yokogawa CSU-X1 spinning-disk unit, and two Hamamatsu C9100-13 EM-CCD cameras, or an instant structured illumination microscope (Visitech iSIM) using an Olympus IX73 body, an Olympus UPLSAPO40XS objective and a Hamamatsu Flash 4.0v2 sCMOS camera. Z stacks composing 30 slices of 1 micron each were used for imaging of compressed embryos, and 36 slices of 2 microns each for uncompressed embryos. Embryos were imaged every 75 s (lineaging) or between 2–5 min (non-lineaging), sufficient time was allowed to enable visualization of terminal phenotypes for easier selection of mutants. Imaging exposure time was 150 ms per slice, with 568 nm laser exposure every timepoint and 488 nm laser exposure every 5 min regardless of imaging frequency.

## Image analysis

Visual analysis of embryos was done using Fiji software (*Schindelin et al., 2012*). In *Figure 1*, Visual tracking of involuting chains was done with Fiji. In *Figure 2b*, to determine degree of involution, we measured the distance between the leading edge of involuting neurons as visible with *cnd1p*::RFP marker in Fiji, identified as the edge of the ventro-lateral *cnd1+* chunk closest to the midline. In *Figure 2c*, to measure degree of pharynx retraction, distance to the anterior tip of the fully retracted pharynx from the anterior tip of the embryo was also measured manually on Fiji. *Figure 2b and c* were both measured in the same 12 embryos. Initial times were identified according to when *cnd1+* ventro-lateral neurons begin moving to the midline; second timepoints were chosen based on when these neurons were the closest toward each other around the midline.

In *Figure 3c*, HMR-1:GFP expression at both the basal pharyngeal interface and the apical side of the pharynx (both identified by eye according to location on the pharynx) were quantified by creating a four slice MAX projection around each location and subsequently drawing an ROI and quantifying fluorescence at three timepoints (before, during, and after pharynx retraction), and normalized to the background fluorescence. Pharyngeal cytoplasmic region was measured as a negative control. This was done across n = 5 embryos. Timepoints were chosen according to pharyngeal shape and retraction distance, with approximately 10 min or less estimated real-timing variance among embryos in a given temporal group.

In *Figure 4c*, HMR-1 measurement within the neuronal tissue mass, a random set of five internal membrane areas were selected and drawn based on *cnd-1* marker expression in the ventro-lateral portion of the embryo. *Cnd-1* labels the majority of non-interface ventro-lateral involuting neurons. ROIs for each membrane segment were drawn while blinded to HMR-1 expression. These boundaries represent the membranes between *cnd-1+* follower cells with a variety of terminal fates including, in order to create a representative sample of involuting neurons. This was repeated for six embryos. None of these regions overlap with the ventral interface hmr-1 patch, as *cnd-1* does not label any interface neurons. Also, only boundaries between two *cnd-1+* cells were selected.

To demonstrate that HMR-1is genuinely enriched on the internal membranes of the cohesive neuron mass we contrast these measurements with background HMR-1 levels in the membrane of neuronal tissue that is not part of the cohesive mass of followers. The amphids sensory neurons are adjacent to the cohesive mass, but do not move w@Sandhiyaith them, and appear to have lower HMR-1 expression. Again five internal membrane regions within the amphid group were drawn

based on *cnd-1* expression and HMR-1 expression levels were measured in six embryos. As further control to show HMR-1 is elevated on the membrane boundary, adjacent cytoplasmic regions were selected and quantified for each measured edge.

Timepoints were chosen halfway through involution. This was according to pharyngeal retraction distance (~50% of total retraction) and *cnd-1+* neuron involution (~50% of distance toward midline from starting location), 5 Z slices were chosen around the ROI for both amphid and involuting neurons separately, and a MAX projection was created in order to do a 2D quantification across multiple Z planes.

## Statistical analysis

Statistical tests used were as follows: *Figure 2b,c*: one-tailed t-test, n = 12 embryos. *Figure 2f*: one-tailed t-test, n = 3 embryos. *Figure 3c*: two tailed t-test, n = 5 embryos. *Figure 3f*: one-tailed t-test, n = 3 embryos. *Figure 4c*: two tailed t-test, n = 6 embryos, 5 cell boundaries per embryo. *Figure 4e*: Paired t-test, n = 3 embryos.

## Computational cell motion analysis

For automated tracking and rendering of cell movement paths, StarryNite software was used to segment RFP tagged nuclei (*Santella et al., 2010*) and AceTree was used to edit cells of interest to assure successful tracking (*Boyle et al., 2006*). MATLAB was used for all visualization and analysis on single identified cells, including renderings of cell movement patterns, cell displacement analysis, and motion path correlation analysis.

Cell Motion: Nuclear positions for each desired cell were extracted over the selected time window. When cell divisions occurred during this window desired cells and their parents were considered equivalent and their positions concatenated. Cell position over time was smoothed using the MATLAB smoothdata function which computes a moving window average at an automatically selected scale (see *Figure 2d and e*, *Figure 3e and f*, *Figure 4d and e*).

Correlation Analysis: Positions were differenced and correlation of 3d directional velocity over all timepoints was computed for all pairs of cells (see *Figure 4d*).

## Cohesive neuron screen

A screen was performed for neurons that maintain close contact with the pharynx over time. Based on nuclear positions a Voronoi diagram approximation of cell–cell contacts was computed. The total of neuron-pharynx contacts over time was computed from this model, using only pharyngeal neighbors from the first timepoint. A threshold of 75% indicating a neuron with near constant contact with at least one pharynx cell was established and used to determine a set of leader neurons. The SMDD neurons, the most highly cohesive left/right neuron pair in the ventral patch of neurons, were selected for use in further computational analysis.

## WormGUIDES analysis

WormGUIDES is a tool which enables visualization of fine spatiotemporal analysis of selected cell movement patterns in the *C. elegans* embryo via an adjustable 3D rendering (*Santella et al., 2015*). Involuting neuronal nuclei were visualized next to pharynx and hypodermal surface models, including visualizing the movement of these features overtime created using the Fiji temporal max projection tool.

## Acknowledgements

We thank Dr. Shyr-Shea Chang and Braden Katzman for assistance with experiments and quantification, and all Bao lab members for general help and feedback. We thank Drs. Hari Shroff, Jeremy Dittman, Pavak Shah, and Srivarsha Rajshekar for comments on the manuscript. Some strains were provided by the CGC, which is funded by NIH (P40 OD010440). This work was partly supported by NIH grants (R01 GM097576 and R24 OD016474) to ZB and a Core Grant to MSKCC (P30 CA008748). Research in the DAC-R lab was supported by NIH grant No. R24-OD016474 and by an HHMI Scholar Award. MWM was supported by NIH by F32-NS098616. AS was supported by grant

2019–198110 (5022) from the Chan Zuckerberg Initiative and the Silicon Valley Community Foundation.

## Additional information

### Funding

| Funder | Grant reference number | Author |
|---|---|---|
| National Institute of General Medical Sciences | R01 GM097576 | Zhirong Bao |
| Office of the Director | R24 OD016474 | Daniel A Colón-Ramos Zhirong Bao |
| National Cancer Institute | P30 CA008748 | Zhirong Bao |
| Chan Zuckerberg Initiative | 2019-198110 (5022) | Anthony Santella |
| Howard Hughes Medical Institute | HHMI Scholar Award | Daniel A Colón-Ramos |
| National Institute of Neurological Disorders and Stroke | F32-NS098616 | Mark W Moyle |

The funders had no role in study design, data collection and interpretation, or the decision to submit the work for publication.

### Author contributions

Kristopher M Barnes, Conceptualization, Data curation, Formal analysis, Supervision, Validation, Investigation, Visualization, Methodology, Writing - original draft, Project administration, Writing - review and editing; Li Fan, Investigation, Microscope Images; Mark W Moyle, Resources; Christopher A Brittin, Yichi Xu, Investigation; Daniel A Colón-Ramos, Writing - review and editing; Anthony Santella, Conceptualization, Resources, Data curation, Software, Formal analysis, Funding acquisition, Validation, Investigation, Visualization, Methodology; Zhirong Bao, Conceptualization, Resources, Data curation, Supervision, Funding acquisition, Validation, Investigation, Project administration, Writing - review and editing

### Author ORCIDs

Kristopher M Barnes (iD) https://orcid.org/0000-0003-2546-9529
Li Fan (iD) http://orcid.org/0000-0003-1780-6919
Christopher A Brittin (iD) https://orcid.org/0000-0002-1143-554X
Daniel A Colón-Ramos (iD) http://orcid.org/0000-0003-0223-7717
Zhirong Bao (iD) https://orcid.org/0000-0002-2201-2745

### Decision letter and Author response

Decision letter https://doi.org/10.7554/eLife.58626.sa1
Author response https://doi.org/10.7554/eLife.58626.sa2

## Additional files

### Supplementary files

• Supplementary file 1. List of neurons used in figure panels identities of neurons used in WormGUIDES renderings and visualizations derived from WormGUIDES data, with corresponding figures listed.

• Transparent reporting form

### Data availability

All data generated or analysed during this study are included in the manuscript and supporting files. Source data files have been provided for Figures 1a,c, 2b,c,e, 3c,f, and 4c and e.

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
