## [Decision Letter]

**Acceptance summary:**

This study provides intriguing new insights into tissue morphogenesis and describes molecular mechanisms underlying this process.

**Decision letter after peer review:**

Thank you for submitting your article "Cadherin Preserves Cohesion Across Involuting Tissues During *C. elegans* Neurulation" for consideration by *eLife*. Your article has been positively reviewed by three peer reviewers, one of whom is a member of our Board of Reviewing Editors, and the evaluation has been overseen by Piali Sengupta as the Senior Editor. The following individual involved in review of your submission has agreed to reveal their identity: Andrew D Chisholm (Reviewer #2).

The reviewers have discussed the reviews with one another and the Reviewing Editor has drafted this decision to help you prepare a revised submission.

As you will see below, each of the reviewers was very positive both about the importance of the problem and the overall quality of your data – but they all point out a few issues that you can address with some textual revisions (with the possible exception of the first comment by reviewer #3; we hope you may already have some data on this available). We are looking forward to seeing a properly revised version of the paper.

Reviewer #1:

Elucidating conserved molecular mechanisms that contribute to coordinated tissue movement during embryogenesis, ultimately resulting in the formation of centralized nervous system across phyla, is of major interest in developmental neuroscience. Details of how this happens in invertebrates are lacking. Prior studies suggest that various families of cell adhesion molecules such as cadherins are broadly expressed in multiple tissue types during this point in embryogenesis, suggesting that they could contribute to nervous system centralization.

In this manuscript, the authors use 3D time-lapse imaging and cell lineage tracing to show that the *C. elegans* nervous system is internalized via coordinated movement of the retracting pharynx and the neuroectoderm. They show that inter and intra-tissue cohesion is mediated by hmr-1, a homolog of vertebrate classical cadherin. HMR-1 transiently localizes at the interface between the pharynx and neurons, as well as between neurons, at the time of involution. The authors show that animals with a loss of function mutation in hmr-1 have reduced and disorganized involution of neurons although pharynx retraction is unaffected. The present study proposes that nervous system involution in *C. elegans* is analogous to neurulation in vertebrates based on conserved cellular events and shared gene expression, favoring the hypothesis that the centralization of the nervous system/origin of the brain, likely occurred once in evolution.

This manuscript adequately shows the role of hmr-1 in internalization of the *C. elegans* nervous system and explicates novel and exciting details to establish the similarity of *C. elegans* nervous system involution to mammalian neurulation. The paper is suitable for publication in *eLife* after addressing the points below to strengthen the support for their conclusions.

Essential revisions:

1) It is not clear what kind of an allele the authors use. There does not appear to be any molecular information about the zu248 and this needs to be remedied. Is it a nonsense/putative null? Does it affect both splice forms or just one? The authors have to provide this information and if they do not have it, they need to sequence the allele. Along those lines, I'm curious to hear why the authors did not analyze existing mutant alleles (available at the CGC) that are specific to the longer isoform, gk3258 (a deletion allele that introduces frameshift) and a missense mutation from the million mutant project. Both alleles appear to *not* cause the lethality associated with the canonical zu289 allele. Do these alleles fail to show any of the phenotypes shown for the zu248 allele? This is not a pure bookkeeping question, but it relates specifically to the notion, discussed by the authors, that it is the longer isoform that more resembles the N-Cadherin. Meaning, their molecular homology argument may be undermined and/or confirmed by examing an "N-Cadherin" (i.e. hmr-1b)-specific allele(s).

2) The expression/localization analysis of hmr-1 is a little unsatisfactory. The authors state that they used an hmr-1b protein fusion to the "hmr-1" promoter. However, it's explicitly known that with its two different isforms, the hmr-1 locus has TWO promoters. Which was used? Using either the 5' region of the hmr-1a or the hmr-1b isoform alone, is prone to yield an incomplete expression pattern. What can the authors say about that?

Reviewer #2:

Barnes et al., investigate morphogenetic events underlying the arrangement of the nervous system in *C. elegans*. Using automatic tracking they find that neuronal cell bodies undergo coordinated involution. This is likely to be driven by force generation in an internal organ (pharynx) that undergoes apical constriction. Expression studies and genetic loss of function tests support a role for the cadherin HMR-1 in this process, linking the pharynx shape change with neural and epidermal movements. The authors speculate that this process is homologous to neurulation in vertebrates.

This is a well written and technically very convincing set of findings. The description of neuron involution and the role of cadherin is novel and will be of interest to *eLife* readership. The main caveat would be whether the analysis has been taken to the level that would generally be expected in this journal. The main set of data that seem lacking are tissue specific tests of the requirement of HMR-1. Those experiments would perhaps be challenging but would take the story beyond a very detailed description of a hmr-1 mutant phenotype (of a single allele). The authors’ claims in the Abstract to have found roles for 'localized HMR-1' would be strengthened if the requirement for localized HMR-1 had been directly tested. If such experiments are not possible, the claims should be toned down to state that.

Reviewer #3:

Summary:

This manuscript investigates involution of anterior neuroblasts in the *C. elegans* embryo using a combination of lineage analysis, microscopy, and analysis of mutants. The authors present evidence that HMR-1/cadherin is required for the integrated morphogenesis of tissues derived from multiple germ layers. The authors advance our understanding of how HMR-1/cadherin facilitates involution of neurons in the anterior by identifying a supercellular accumulation of HMR-1 at the interface between neurons and the pharyngeal primordium.

The imaging is high quality, the use of WormGuides and other existing lineage tracing data is creative, the use of "tadpole" tracings is visually effective, and the basic story seems clear. Two main issues should be addressed to improve the manuscript, along with some minor issues. If these issues can be addressed, then this is a valuable contribution.

Essential revisions:

1) Statistical analysis of quantification of HMR-1 accumulation: Figure 3C seems to depict a single embryo. In the absence of aggregate data, it is really impossible to assess these data. Similarly, Figure 4B is suggestive, but in the absence of quantification, it is hard to evaluate whether HMR-1 really does accumulate on a consistent basis between anterior neuroblasts in a statistically significant manner.

2) Homology discussion: The discussion of homology is deeply problematic. This study cannot distinguish between "deep homology" involving convergent evolution using conserved molecular cassettes, versus "real" homology via shared common ancestors. The idea that there was an ancestral bilaterian possessing a single tissue attached to the neuroectoderm that gradually evolved independently in the deuterostome and protostome lineages is interesting but speculative. In Figure 5, the authors try to compare worms to several chordates. If a pan-metazoan conservation of process is being proposed, however, then it would be valuable to show another protostome (annelid? arthropod?). I recommend serious overhaul of this section, including trimming it to remove excessive speculation.

---

## [Author Response]

As you will see below, each of the reviewers was very positive both about the importance of the problem and the overall quality of your data – but they all point out a few issues that you can address with some textual revisions (with the possible exception of the first comment by reviewer #3; we hope you may already have some data on this available). We are looking forward to seeing a properly revised version of the paper.

We thank the reviewers for their thoughtful and constructive feedback, especially Dr. Chisholm for making the review process more transparent. As detailed below, we have addressed the suggestions with both textual revisions and additional data analysis. In terms of textual revisions, we revised the introduction on vertebrate neurulation to better reflect the nuances as well as the discussion on the evolutionary implication. In terms of additional data analysis, we added statistics in Figure 3 and Figure 4 as suggested. We also highlighted the major changes in the manuscript.

Reviewer #1:Essential revisions:1) It is not clear what kind of an allele the authors use. There does not appear to be any molecular information about the zu248 and this needs to be remedied. Is it a nonsense/putative null? Does it affect both splice forms or just one? The authors have to provide this information and if they do not have it, they need to sequence the allele. Along those lines, I'm curious to hear why the authors did not analyze existing mutant alleles (available at the CGC) that are specific to the longer isoform, gk3258 (a deletion allele that introduces frameshift) and a missense mutation from the million mutant project. Both alleles appear to not cause the lethality associated with the canonical zu289 allele. Do these alleles fail to show any of the phenotypes shown for the zu248 allele? This is not a pure bookkeeping question, but it relates specifically to the notion, discussed by the authors, that it is the longer isoform that more resembles the N-Cadherin. Meaning, their molecular homology argument may be undermined and/or confirmed by examing an "N-Cadherin" (i.e. hmr-1b)-specific allele(s).

*zu248* is a loss of function allele that was reported together with *zu389* in the paper that defined the *hmr-1* gene in *C. elegans* (Costa et al., 1998). *zu389* contains a nonsense mutation and is the most widely used. However, while using JJ1079 [hmr-1(zu389)/lin-11(n566) unc-75(e950) I], we lost the mutation twice during strain maintenance (after two separate requests from CGC). Worrying about unknown complications in the genetic background, we switched to using JJ1142 [hmr-1(zu248) I; zuEx2]. The molecular lesion of *zu248* has not been defined, but according to Costa et al., *zu248* has comparable phenotypes as *zu389*.

We thank the reviewer for pointing out the additional alleles. We also agree that sorting out the requirement of the long and short isoforms is a meaningful step towards a better understanding of the molecular mechanisms, together with some suggestions below. However, we feel that these deserve a systematic set of experiments of their own rights and therefore better suited for the next step. We recognize that it is premature to discuss the N- vs E-Cadherin without these experiments and removed it from the Discussion section.

2) The expression/localization analysis of hmr-1 is a little unsatisfactory. The authors state that they used an hmr-1b protein fusion to the "hmr-1" promoter. However, it's explicitly known that with its two different isforms, the hmr-1 locus has TWO promoters. Which was used? Using either the 5' region of the hmr-1a or the hmr-1b isoform alone, is prone to yield an incomplete expression pattern. What can the authors say about that?

The reporter we used is *xnIs96* [hmr-1p::hmr-1::GFP::unc-54 3'UTR] reported in (Achilleos et al., 2010). To quote Achilleos et al., “hmr-1::gfp was constructed by replacing the hmr-1 genomic sequence in plasmid pW02-21 (Broadbent and Pettitt, 2002) with hmr-1 cDNA containing introns 2-4, inserting an ApaI site before the stop codon, and cloning gfp plus the unc-54 3' UTR into this site. HMR-1GFP produced from the hmr-1::gfp transgene xnIs96 is localized similarly to endogenous HMR-1 as detected by immunostaining, and rescues the strict embryonic lethality of hmr-1(zu389) mutants [1064 of 1485 (72%) were viable].” Our reference to *hmr-1b* in the text is a typo and we apologize for the oversight.

Reviewer #2:This is a well written and technically very convincing set of findings. The description of neuron involution and the role of cadherin is novel and will be of interest to eLife readership. The main caveat would be whether the analysis has been taken to the level that would generally be expected in this journal. The main set of data that seem lacking are tissue specific tests of the requirement of HMR-1. Those experiments would perhaps be challenging but would take the story beyond a very detailed description of a hmr-1 mutant phenotype (of a single allele). The authors’ claims in the Abstract to have found roles for 'localized HMR-1' would be strengthened if the requirement for localized HMR-1 had been directly tested. If such experiments are not possible, the claims should be toned down to state that.

We agree that tissue specific experiments would be the ultimate test, but as mentioned above, we do not have good promoters to target the interface neurons. We revised the Abstract as suggested to tone down the statement.

Reviewer #3:Essential revisions:1) Statistical analysis of quantification of HMR-1 accumulation: Figure 3C seems to depict a single embryo. In the absence of aggregate data, it is really impossible to assess these data. Similarly, Figure 4B is suggestive, but in the absence of quantification, it is hard to evaluate whether HMR-1 really does accumulate on a consistent basis between anterior neuroblasts in a statistically significant manner.

For Figure 3C, we quantified additional embryos (n=5) and revised the panel. For Figure 4B, we made quantitative measurements (6 embryos and 5 neurons/embryo) along with controls, and added a new panel (new Figure 4C).

2) Homology discussion: The discussion of homology is deeply problematic. This study cannot distinguish between "deep homology" involving convergent evolution using conserved molecular cassettes, versus "real" homology via shared common ancestors. The idea that there was an ancestral bilaterian possessing a single tissue attached to the neuroectoderm that gradually evolved independently in the deuterostome and protostome lineages is interesting but speculative. In Figure 5, the authors try to compare worms to several chordates. If a pan-metazoan conservation of process is being proposed, however, then it would be valuable to show another protostome (annelid? arthropod?). I recommend serious overhaul of this section, including trimming it to remove excessive speculation.

We made significant revision on this part of the manuscript as suggested. In the Results section, we reduced the section so that it only describes the observed similarities (the facts). In the Discussion section, we turned the speculations into what could be examined in both deuterostomes and protostomes to test homology vs convergent evolution.